# Pyrolysis Kinetics of Hydrochars Produced from Brewer's Spent Grains

**Maciej P. Olszewski** [1],* **, Pablo J. Arauzo** [1]**, Przemyslaw A. Maziarka** [2] **, Frederik Ronsse** [2] **and Andrea Kruse** [1]

[1]  Department of Conversion Technologies of Biobased Resources, Institute of Agricultural Engineering, University of Hohenheim, Garbenstrasse 9, 70599 Stuttgart, Germany

[2]  Department of Green Chemistry and Technology, Faculty of Bioscience Engineering, Ghent University, Coupure Links 653, 9000 Gent, Belgium

*  Correspondence: maciej.olszewski@uni-hohenheim.de; Tel.: +49-711-459-24737

**Abstract:** The current market situation shows that large quantities of the brewer's spent grains (BSG)—the leftovers from the beer productions—are not fully utilized as cattle feed. The untapped BSG is a promising feedstock for cheap and environmentally friendly production of carbonaceous materials in thermochemical processes like hydrothermal carbonization (HTC) or pyrolysis. The use of a singular process results in the production of inappropriate material (HTC) or insufficient economic feasibility (pyrolysis), which hinders their application on a larger scale. The coupling of both processes can create synergies and allow the mentioned obstacles to be overcome. To investigate the possibility of coupling both processes, we analyzed the thermal degradation of raw BSG and BSG-derived hydrochars and assessed the solid material yield from the singular as well as the coupled processes. This publication reports the non-isothermal kinetic parameters of pyrolytic degradation of BSG and derived hydrochars produced in three different conditions (temperature-retention time). It also contains a summary of their pyrolytic char yield at four different temperatures. The obtained KAS (Kissinger–Akahira–Sunose) average activation energy was 285, 147, 170, and 188 kJ mol$^{-1}$ for BSG, HTC-180-4, HTC-220-2, and HTC-220-4, respectively. The pyrochar yield for all hydrochar cases was significantly higher than for BSG, and it increased with the severity of the HTC's conditions. The results reveal synergies resulting from coupling both processes, both in the yield and the reduction of the thermal load of the conversion process. According to these promising results, the coupling of both conversion processes can be beneficial. Nevertheless, drying and overall energy efficiency, as well as larger scale assessment, still need to be conducted to fully confirm the concept.

**Keywords:** hydrothermal carbonization; pyrolysis; kinetics; hydrochar; biomass; spent grain; lignocellulose; waste valorization

## 1. Introduction

Beer has been classified in the third position of the most popular drinks worldwide, after water and tea [1]. The global production of beer reached 196 billion ($10^9$) liters in 2016 [1,2]. Vast amounts of residues are generated during the production of beer, mostly in the form of brewer's spent grains (BSG). Rough estimation states that production of one liter of beer results in the formation of 0.14–0.20 kg of wet BSG (20 wt.%–30 wt.% dry matter). Taking into account the worldwide beer production, it gives an approximate production of 27–39 million ($10^6$) metric ton wet BSG per year [3,4].

The composition of BSG may vary depending on the barley species and brewing technology used. In general, it can be stated that BSG (wt.% dry matter) consists of hemicellulose (21.8 wt.%–40.2 wt.%), cellulose (12.0 wt.%–12.54 wt.%), lignin (4.0 wt.%–27.8 wt.%), lipids (3.9 wt.%–13.3 wt.%) and proteins

(14.2 wt.%–26.7 wt.%) [5,6]. Due to the high protein and fiber content, the most common application of BSG is as a low-cost cattle feed. However, the vast quantities of BSG, produced by the largest breweries can only be partially utilized in this way. The freshly produced BSG after mechanical dewatering has a high moisture content (i.e., 70 wt.%–80 wt.% moisture on a wet feedstock basis), which leads to a significant increase in the transportation cost with distance from a brewery. As a result, use of BSG as cattle feed is only economically feasible for farmers located in the near vicinity of the breweries. Moreover, wet raw biomasses have a short period of storage, as they constitute a favorable environment for the fast growth of microbes (molds and bacteria). Feeding animals with contaminated feed can lead to diseases, in the most severe cases even to the death of the whole heard. As such, BSG should be dried to a moisture level below 10 wt.% to minimize microbiological decay [3,4,7,8] and to prolong the storage time. Nevertheless, drying of the process leftovers is not convergent to the work of breweries nor gives tangible benefits to them, so the costs of drying are usually on the side of the farmers. As a consequence, the number of potential BSG utilizers is limited. Based on the abovementioned statements, there is an excess amount of the BSG available on the market that can be utilized or valorized in other ways than animal feed while simultaneously not providing economic competition to the latter application. From this perspective, brewer's spent grains are a valuable stream of lignocellulosic waste biomass, which may be valorized through biotechnological processes (i.e., anaerobic digestion to methane) or via thermochemical conversion processes (i.e., pyrolysis or hydrothermal carbonization).

Drying of raw lignocellulosic biomass is a high energy demanding process due to the porosity and high hydrophilicity (and thus, large amounts of bound water) of its structural components. Therefore, within valorization through thermochemical processes, the focus should be on those processes in which deep drying before the conversion is not necessary. Such solutions will be favorable in terms of overall energy efficiency. In case of processing through pyrolysis, the biomass has to be dried down to at least 30 wt.% moisture content, but even lower moisture levels are considered favorable in pyrolysis. As a consequence, the operational costs in pyrolysis sharply increase for very wet biomass feedstocks, like BSG (Figure 1, Scenario A). The most promising thermochemical conversion option is the hydrothermal carbonization process (HTC). The HTC process, in principle, converts the biomass into a solid product, enriched in carbon, using liquid water under subcritical conditions as the reaction medium. HTC operates at elevated temperatures (180–260 °C) and pressures above the corresponding water vapor pressure to keep the medium in the liquid phase [7,9–14]. Since the reaction medium is based on water, the high moisture content of the feedstock is favorable in such a process. HTC allows expanding the potential range of biomass applications for bioenergy purposes since the drying step of raw biomass is avoided. Through HTC, it is possible to convert feedstocks, which are considered troublesome for processes like pyrolysis or gasification. These aforementioned feedstocks include very wet biomasses, containing between 70 wt.% and 90 wt.% of water, examples thereof are bio-waste streams (wastewater sludge, bio-refinery digestate, pulp, and paper sludge) and food production leftovers (brewer's spent grains, sugar beet bagasse, fruit pomace) [7,10,14–17]. The solid end-product of HTC often termed hydrochar, is a carbonized material, and hence, its surface is more hydrophobic than that of the initial feedstock [9]. Consequently, the hydrochars are more suitable for mechanical dewatering, which can reduce the moisture content to approximately 50 wt.%, depending on the used technology [18]. The most significant advantage of the mechanical dewatering in comparison to thermal drying is its significantly lower operational cost. Another benefit of HTC is the reduction of the amount of the material which has to be dried, inversely proportional to HTC yield. HTC results in a more economically and energetically efficient process, due to reduced drying stages and saving the heat necessary to evaporate water from the hydrochars.

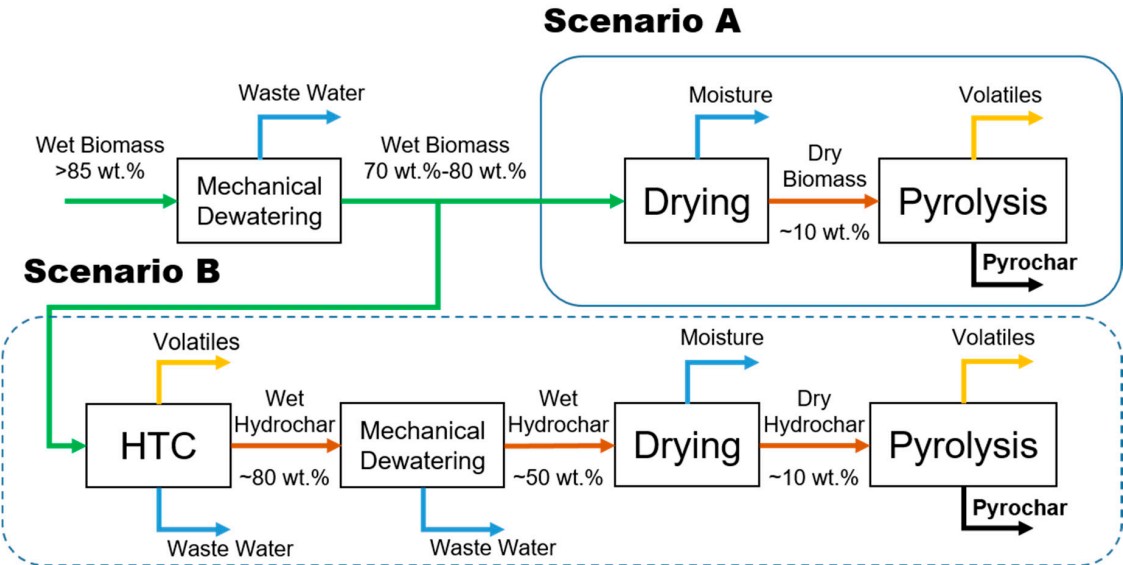

**Figure 1.** Simplified process flow diagram; Scenario (**A**) conventional pyrolysis, Scenario (**B**) the integration of hydrothermal carbonization (HTC) with pyrolysis.

Hydrochar is stable and can be stored for a prolonged time. It has a wide range of applications, for example as a solid fuel in electricity generation in combined heat and power (CHP) plants or as a reductant in the metallurgical industry. Unfortunately, hydrochars may not fully be suitable for more technologically advanced applications. For instance, one potential use for hydrochar is as a carbon-negative soil amendment or peat replacement in plant growth substrates. However, recent reports state that fresh hydrochars may have a toxic effect on plant growth, therefore fresh hydrochars show limited applications as a soil amendment [19]. One way to extend the range of the hydrochars applications is a secondary treatment processes. One of the consecutive processes of the hydrochar refining or upgrading can be a pyrolysis process as is shown in Figure 1, Scenario B. Pyrolysis, because of its operational simplicity and its relatively low operational costs, suits best the purpose of upgrading hydrochars. Pyrolysis can be applied to increase the hydrochar carbon content, surface area, and decrease its phytotoxicity. As a result, the final, more refined char can be used in a much broader range of applications, like soil amendment, production of activated carbons, catalysts and catalyst supports, and as a carbon-rich material for supercapacitors or carbon electrodes [18–20].

Parameters describing the kinetics (activation energy and pre-exponential factor A) are instrumental in understanding the pyrolysis process. With this information, it is possible to get an overview of the progression of hydrochar pyrolysis. Using a model-free method, it is possible to calculate this kinetic data without accurate knowledge about the reaction mechanisms that occur in complex processes such as pyrolysis. The kinetic parameters allow for the establishment of base models for the development and optimization of chemical reactors (through computational fluid dynamics (CFD) modeling), providing necessary knowledge related to the decomposition of the material at elevated temperatures with time. It can also be used for consecutive scaling up as well as further process optimization (for example, in process simulation software like Aspen Plus) to maximize the yields and minimize the energy consumption of the process. The fundamental study of global pyrolysis kinetics of hydrochars is necessary to understand their thermal degradation, which is the first step to be able to assess the feasibility of the proposed novel route of the combined BSG processing. For this purpose and in this study, hydrochars were produced in three different process conditions: (i) 180 °C, 4 h residence time; (ii) 220 °C, 2 h; and (iii) 220 °C, 4 h to investigate the effect of HTC process variables on the hydrochars decomposition behavior during the subsequent pyrolysis process. The kinetic parameters (activation energy and pre-exponential factor) were estimated based on Kissinger–Akahira–Sunose (KAS) method. The initial biomass and hydrochars were pyrolyzed at 300, 500, 700, and 900 °C using a

TGA instrument to compare the proposed Scenarios A and B (Figure 1). The proximate and ultimate analysis of all obtained pyrolysis chars were carried out and compared with the initial materials (for the latter chars, the term pyrochar will be used further throughout the manuscript in order to distinguish from hydrochar).

## 2. Results and Discussion

### 2.1. Feedstocks Characteristics

The results of proximate, elemental analysis, and higher heating value (HHV) determination of three hydrochars (HTC-180-4, HTC-220-2, and HTC-220-4) produced at different process conditions are shown in Table 1. The hydrochar yields were 67.5 wt.%, 58.0 wt.%, 55.0 wt.% (dry basis; db), respectively [21]. Summarized results from the proximate and ultimate analysis of BSG and hydrochars are also presented in Table 1. Analyzed materials showed moisture contents ranging from 3.25 wt.%–4.06 wt.%, despite earlier drying. A slight decrease in the hydrochar moisture content is due to the increasing hydrophobicity with increasing HTC process temperature [10]. The ash content of the initial biomass was 4.32 wt.% (db), and moreover, the ash content raised slightly under more severe conditions up to 4.83 wt.% (db) for hydrochars produced at 220 °C, and 4 h residence time. The ash content of hydrochars was expected to be higher, based on the hydrothermal carbonization yield. Obtained results indicate that part of the inorganic constituents was removed during the process, e.g., by dissolving the sodium and potassium salts (carbonates, chlorides, and phosphates) [10,17,22,23]. The thermochemical conversion of biomass reduces volatile matter (VM), as well as increases fixed carbon (FC) content in its final product by changing the chemical structure of the converted solid matter. Here, the hydrothermal conversion of brewer's spent grains decreased VM from 79.22 wt.%–63.93 wt.% and increased FC content from 16.46 wt.%–31.25 wt.%. Furthermore, hydrochars had higher carbon and lower oxygen contents than the original biomass caused by dehydration, decarboxylation, condensation, aromatization, and polymerization reactions occurring during HTC [9]. The conversion resulted in a noticeable increase in the estimated HHV of the hydrochars (~29.5–31 MJ kg$^{-1}$), comparable to lignite (on dry ash free basis), and much higher than the initial biomass HHV (23.59 MJ kg$^{-1}$ dry ash-free (daf) basis). In addition, BSG has a high protein content [5,6]. Results show that the nitrogen content from the raw biomass (4.89 wt.%, daf) stayed at the same level in hydrochars (4.67 wt.% daf), despite the mass loss upon hydrochar production in the conversion process. It leads to a strong conclusion that a part of the feedstock nitrogen had to be transferred into the process water in the form of organic compounds [24].

**Table 1.** Proximate and ultimate analysis of brewer's spent grains and its hydrochars.

| Analysis | BSG | | HTC-180-4 | | HTC-220-2 | | HTC-220-4 | |
|---|---|---|---|---|---|---|---|---|
| Moisture, wt.% | 3.75 | ± 0.65 | 4.06 | ± 0.41 | 3.42 | ± 0.75 | 3.25 | ± 0.83 |
| **Proximate analysis, db, wt.%** | | | | | | | | |
| Ash | 4.32 | ± 0.05 | 4.40 | ± 0.05 | 4.40 | ± 0.13 | 4.83 | ± 0.13 |
| Volatile Matter | 79.22 | ± 0.07 | 70.59 | ± 0.17 | 68.06 | ± 0.57 | 63.93 | ± 1.03 |
| Fixed Carbon | 16.46 | ± 0.76 | 25.01 | ± 0.24 | 27.54 | ± 0.63 | 31.25 | ± 0.57 |
| **Ultimate analysis, daf, wt.%** | | | | | | | | |
| C | 53.50 | ± 0.40 | 66.29 | ± 0.33 | 68.82 | ± 0.75 | 70.17 | ± 0.77 |
| H | 7.27 | ± 0.09 | 7.39 | ± 0.04 | 7.62 | ± 0.05 | 7.21 | ± 0.06 |
| N | 4.89 | ± 0.27 | 4.54 | ± 0.06 | 4.61 | ± 0.06 | 4.67 | ± 0.14 |
| S | 0.30 | ± 0.02 | 0.46 | ± 0.01 | 0.45 | ± 0.01 | 0.45 | ± 0.02 |
| O | 34.04 | ± 0.80 | 21.33 | ± 0.45 | 18.51 | ± 0.65 | 17.51 | ± 1.00 |
| HHV, daf, MJ kg$^{-1}$ | 23.53 | ± 0.16 | 29.54 | ± 0.11 | 30.89 | ± 0.25 | 31.06 | ± 0.23 |

db—dry basis, daf—dry ash-free basis.



## 2.2. Pyrolysis Char Characteristic

Table 2 presents char yield and proximate and ultimate analysis results for pyrochars. The relative error in all measurements was below 5%. In the materials pyrolyzed in the temperature range between 300 °C and 500 °C, a significant reduction in the volatile matter content was noticed, i.e., from 56.5 wt.%–3.9 wt.% (db) for plain BSG pyrolysis and 40.0 wt.%–51.1 wt.% (db) to 6.2 wt.%–8.8 wt.% (db) for BSG-derived hydrochars pyrolysis. Pyrolysis resulted in significant energy densification when comparing the initial biomass with its derived pyrochar, (e.g., 23.53 MJ kg$^{-1}$ for BSG and 32.22 MJ kg$^{-1}$ for BSG-derived pyrochar produced at 500 °C). However, when hydrochars were used as input to the pyrolysis process, the increase in HHV was not equally high as some energy densification already took place during the HTC process, as discussed in Section 2.1.

During pyrolysis, the increase in elemental carbon was usually contrived with a drop in elemental hydrogen content. The loss of VM was also minimal at the lower temperature in the range of temperatures tested (300 °C). Therefore, the pyrochar yield was the highest (up to 78.5 wt.% for BSG and 85.8 wt.% for HTC-220-4) at 300 °C. Despite this high char yield, the properties of the pyrochars produced at 300 °C were not satisfactory, limiting their potential applications. Pyrochars produced at a relatively low pyrolysis temperature (300 °C) had high VM content which could indicate a low porosity [25] but on the other hand, good reactivity and combustion properties [26,27]. Moreover, the pyrolysis of hydrochars can remove the organic compounds causing phytotoxicity in fresh HTC chars. As a result, the pyrolysis treatment opens up hydrochars application for soil amendment [17,28]. Increasing the temperature to 500–700 °C resulted in more severe thermal degradation and related with it, a stronger decrease in volatile matter content down to 0.3 wt.%–3.8 wt.% (db) as well as a slight increase in the higher heating value up to 31.5–33 MJ kg$^{-1}$. However, these higher pyrolysis temperatures reduced the char yields significantly (22.3 wt.% for biomass and 30.9 wt.%–40.2 wt.% for hydrochars).

The content of volatile matter of the pyrochars produced in this temperature range can indicate their structural change (porosity and specific surface area). The pyrochars produced in the 500–700 °C temperature range should be suitable precursors for activated carbon production via activation with steam or $CO_2$. As a result of increased porosity, the activation gases can easily enter into the internal structure of the material [29]. The pyrolysis temperature range of 700–900 °C characterized a region of the lowest char yields: 22.0 wt.%, 28.3 wt.%, 37.4 wt.%, and 45.8 wt.% (db) for BSG, HTC-180-4, HTC-220-2, and HTC-220-4, respectively. Due to the most severe thermal conditions and the most advanced degradation obtained, pyrochars had the lowest content in volatiles and the highest fixed carbon content of 81 wt.%–89.8 wt.% (db). Such severe conversion conditions result in a slightly lower HHV due to a change in the ratio of organic to the mineral matter contained in these materials as well as the reduction of elemental hydrogen contained into the char. Mochidzuki et al. [30] reported that there is an internal structural change (strong cross-linking and aromatization of the structure) due to the higher carbonization degree (above 700 °C), which results in a reduction of the porosity and increasing electrical conductivity. These chars were carbon-rich materials which have the potential for use as carbon electrodes [31] or as reductant in metallurgical (e.g., silicon and ferrosilicon) industry [32].

The van Krevelen diagram (Figure 2) shows H/C versus O/C atomic ratios for the initial biomass (BSG), hydrochars, and pyrochars produced at 300, 500, 700 and 900 °C. H/C and O/C atomic ratios are the two most commonly used indicators, which represent the carbonization degree of lignocellulosic materials. High values for these atomic ratios are related to untreated or low-carbonized materials such as biopolymers (hemicellulose and cellulose). Lower values of atomic ratios correspond to more carbonized materials like peats, lignite, coals, and anthracite [15]. The H/C ratio (~1.3) and O/C ratio (~0.2) for hydrochars makes them comparable in terms of the carbonization degree with the lignite (brown coal). Increasing the pyrolysis temperature to 900 °C reduces the H/C and O/C ratios significantly into the range of anthracite. Such high temperature of pyrolysis results in the formation of the most condensed and aromatic structure and mimics in this way the natural coalification process which occurred in the Earth's crust for millions of years. As shown in Figure 2, above 500 °C, the

H/C and O/C ratios for pyrochars produced at the same pyrolysis temperature are very similar for all investigated materials.

**Table 2.** Pyrolysis char yield and proximate analysis for brewer's spent grains and hydrochars.

| Pyrolysis Temperature (°C) | Feedstock | Char Yield * | Ash | VM | FC | HHV MJ kg⁻¹ |
|---|---|---|---|---|---|---|
| | | wt.% | | | | |
| | | db | | | | daf |
| 300 | BSG | 78.5 | 5.31 | 56.45 | 38.24 | 27.26 |
| | HTC-180-4 | 79.4 | 5.31 | 51.13 | 43.56 | 29.49 |
| | HTC-220-2 | 81.0 | 5.26 | 43.59 | 51.15 | 30.99 |
| | HTC-220-4 | 85.8 | 5.44 | 40.05 | 54.51 | 31.12 |
| 500 [1] | BSG | 25.9 | 13.08 | 6.89 | 80.03 | 32.22 |
| | HTC-180-4 | 35.8 | 11.78 | 7.50 | 80.72 | 31.59 |
| | HTC-220-2 | 46.2 | 9.21 | 8.83 | 81.96 | 31.11 |
| | HTC-220-4 | 52.0 | 8.98 | 6.21 | 84.80 | 31.88 |
| 700 | BSG | 22.3 | 18.67 | 0.29 | 81.03 | 33.09 |
| | HTC-180-4 | 30.9 | 13.64 | 2.63 | 83.74 | 32.34 |
| | HTC-220-2 | 41.2 | 10.34 | 3.79 | 85.87 | 31.48 |
| | HTC-220-4 | 46.2 | 10.11 | 0.43 | 89.46 | 33.13 |
| 900 | BSG | 22.0 | 18.92 | – | 81.08 | 32.72 |
| | HTC-180-4 | 28.3 | 14.90 | – | 85.10 | 31.94 |
| | HTC-220-2 | 37.4 | 11.39 | – | 88.61 | 30.73 |
| | HTC-220-4 | 45.8 | 10.20 | – | 89.80 | 30.75 |

[1] Data at 500 °C adapted from [33]. * Char yield = $m_{product}/m_{substrate} \times 100\%$, where products are hydrochar and pyrochar, substrates are brewer's spent grains (BSG) or hydrochar in case of hydrochar pyrolysis. db—dry basis; daf—dry ash-free basis.

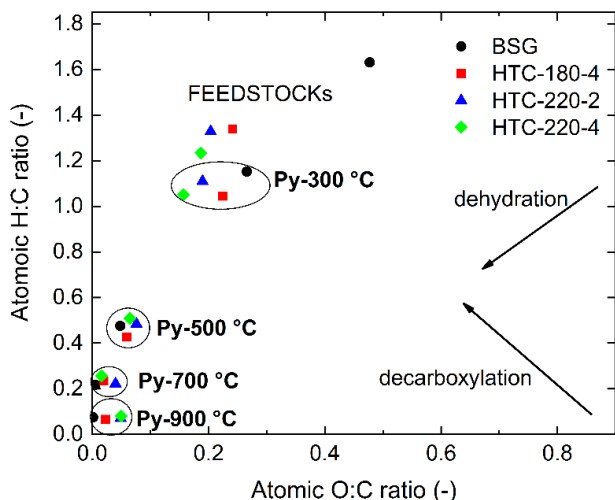

**Figure 2.** Van Krevelen diagram for feedstocks and pyrochars produced at 300, 500, 700, and 900 °C.

*2.3. Analysis of TG–DTG Curves*

The TG (mass loss) curves and DTG (derivative of TG) as a function of temperature showed characteristic tendencies associated with the thermal decomposition of investigated lignocellulosic materials. The exemplary results of the thermogravimetric analysis of BSG and its derived hydrochars, analyzed in TG at 20 °C min⁻¹ are shown in Figure 3A,B. In Table A1 (Appendix A) a summary of parameters in the form of the location of the DTG's peaks with their corresponding temperatures and the final solid residue (char yields) for all investigated samples is presented. The TG curve

for BSG showed a significant discrepancy in comparison with the curves obtained for hydrochars, especially in the initial stage in the temperature range of 200–300 °C. Hydrochar samples began to decompose at lower temperatures than their parent biomass, which indicated lower thermal stability in this temperature range. Such an effect was reflected in the DTG curves, where the peak for BSG (DTG$_1$ ~293 °C) was shifted to the left for the hydrochars (DTG$_{1*}$ at lower temperatures, around 230 °C). Peak DTG$_1$ in biomass is related to the decomposition of the least thermally stable structural biopolymers of lignocellulosic biomass cell wall constituents, i.e., hemicellulose (250–330 °C) [34,35]. During the hydrothermal carbonization process, the hemicellulose is hydrolyzed at temperatures around 180 °C [9]. Therefore, the peak DTG$_{1*}$ may have been related to the decomposition of less stable or extreme side parts of the hydrochar structure (e.g., short-chained polymers) created during polymerization of dissolved molecules during hydrothermal carbonization [16]. On the other hand, the hydrochars were produced under elevated pressure in the liquid environment, and they were not post-treated after production (e.g., washing with water). It leads to the suspicion that their internal structure could be saturated with organic compounds created during the process (e.g., acetic acid and 5-hydroxymethylfurfural), whose decomposition/release may have been visible and overlapped with the peak DTG$_{1*}$ at temperatures lower than the temperatures for hemicellulose degradation. The second characteristic peak DTG$_2$ (~360 °C) is associated mostly with the decomposition of the cellulose (350–420 °C) [36]. However, in the case of BSG, the presence of proteins may have affected the intensity of this peak. The DTG curves (Figure 3B) for biomass and hydrochars obtained at different conditions showed DTG$_2$ peaks at the same temperature. For the hydrochar produced at 180 °C, the peak height was relatively the same as for BSG, but with the increase in the HTC temperature as well as the HTC residence time, the DTG$_2$ peak height dropped significantly. Research published by Kruse et al. [16] and Funke et al. [9] states that cellulose during HTC starts to hydrolyze at a temperature around 200 °C. Therefore, it can be stated that for this study, the cellulose did not hydrolyze at a temperature of 180 °C, and at 220 °C could not undergo complete conversion even with 4 h residence time, due to insufficiently high reaction temperature and reaction time. The last residual peak DTG$_3$ (~420 °C) in case of BSG conversion may have been related to the decomposition of proteins which degraded in the temperature range of 200–500 °C [37]. The DTG$_3$ peak was slightly larger for hydrochars. One of the reasons for such a result could be the degradation of intermediate carbonization products, which were formed during the hydrothermal conversion (i.e., polymerized hydrochar as well as aromatic products of Maillard reactions) [16]. A partial cross-linking of the hydrochar molecules could have been related to the hydrochars higher thermal stability in the temperatures above 200 °C, which were observed from the comparison of TG curves for the hydrochars and the initial biomass. The highest mass loss for all samples was observed up to the temperature around 500 °C. Further increase in the temperature resulted in a flattening of TG and DTG curves indicating insignificant material decomposition above the aforementioned temperature. In spite of that, hydrochars had a higher residual char yield at the end of the TGA measurement, indicating increased thermal stability of the hydrochars. As a result, pyrolysis yield at 900 °C for raw BSG was 19.91 wt.%, and the final residues for hydrochar pyrolysis were 28.30 wt.%, 40.68 wt.%, and 45.75 wt.% for HTC-180-4, HTC-220-2, and HTC-220-4, respectively. Therefore, it can be stated that the HTC process applied before pyrolysis significantly increased the pyrochar yields. Moreover, higher temperature and residence times in hydrothermal carbonization had a significant influence on char yields due to a higher carbonization degree of the hydrochars. It needs to be stressed that, the application of HTC as an additional process prior to pyrolysis (Figure 1, Scenario B) decreased the amount of feedstock entering the pyrolysis process inversely proportional to the hydrothermal carbonization yield. In this way, the material was already pre-processed before pyrolysis, resulting in a lower mass reduction in the final pyrolysis process.

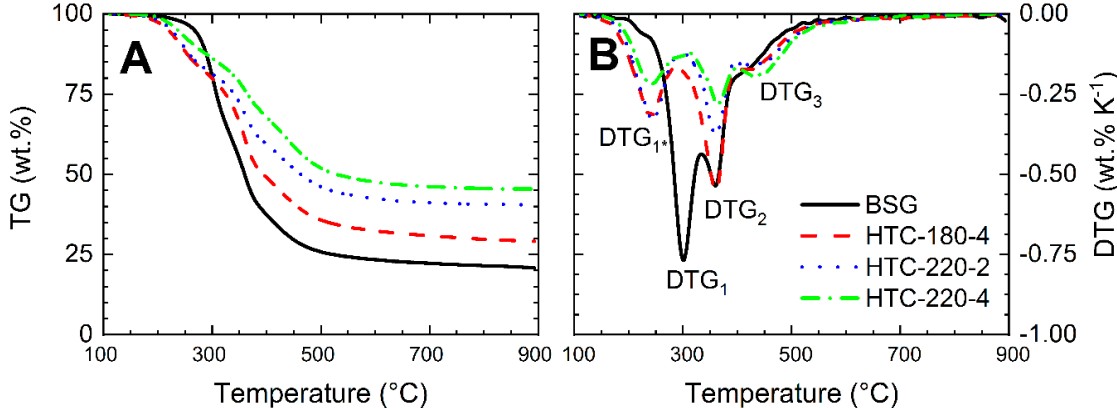

**Figure 3.** (**A**) TG and (**B**) DTG curves for BSG and hydrochars, heating rate 20 °C min$^{-1}$.

During pyrolysis experiments, the influence of the heating rate on the final char yield was observed. In this study, the applied heating rates were within a rather narrow the range (5–40 °C min$^{-1}$), which belongs to the range of slow pyrolysis. However, increasing the heating rate resulted in a slightly higher final char yield, which is contradictory to literature reports [38]. The higher char yield may be caused by a heat transfer limitation together with a significant reduction in residence time (20 min at 40 °C min$^{-1}$ versus 160 min at 5 °C min$^{-1}$). The temperature sensor in the TGA instrument indicates the temperature inside the furnace rather than the exact temperature within the sample. Higher heating rates may result in the sample not being able to heat up so fast within such a short time. As a result, the real temperature of the sample could have been lower than the one indicated by the instrument, which in the end led to a slightly higher char yield.

The literature states that, during pyrolysis, a less steep heating rate increases the residence time of the volatiles inside the particles, due to slower reaction rates [39–41]. Consequently, a lower amount of the evolved vapors and a lower internal pressure in the porous structures of the biomass particles occur. In terms of the transport phenomena, the biomass conversion is a relatively fast process, so the convection is in most cases, the leading transport factor [42,43]. The convection strength and intrinsic gas velocity are dependent on the pressure gradient within the particle. Slower reaction rates/lower instantaneous temperatures in larger biomass particles results in the development of a lower pressure gradient than in case of rapid conversion [44]. Prolongation of the vapor residence time, as well as the increase in their concentration, results in a higher char yield due to secondary char formation (cracking, re-polymerization, and re-condensation reactions) [41,45,46]. This effect is strongly visible when comparing extreme cases, like slow pyrolysis (6–120 °C min$^{-1}$) whose aim is the production of biochar, and flash pyrolysis (>60,000 °C min$^{-1}$) which aims at maximal bio-oil production [47].

Additionally, with an increase in the heating rate, the temperature corresponding to the characteristic DTG peaks was higher (Table A1, Appendix A). This was caused by a change in internal heat transfer as a consequence of the faster heating rate. A similar trend was also observed by other researchers, who investigated the thermogravimetric decomposition of lignocellulosic materials [34,38,48–50].

### 2.4. Pyrolysis Kinetics

Calculation of the kinetic parameters according to the KAS method for BSG and hydrochars was carried out in the temperature range of 105–800 °C, for four heating rates of 5, 10, 20, and 40 °C min$^{-1}$. Linear fit plots for selected conversion degrees are shown in Figure 4. The slope was used to calculate the activation energy for each conversion point. The obtained results are presented in Figure 5.

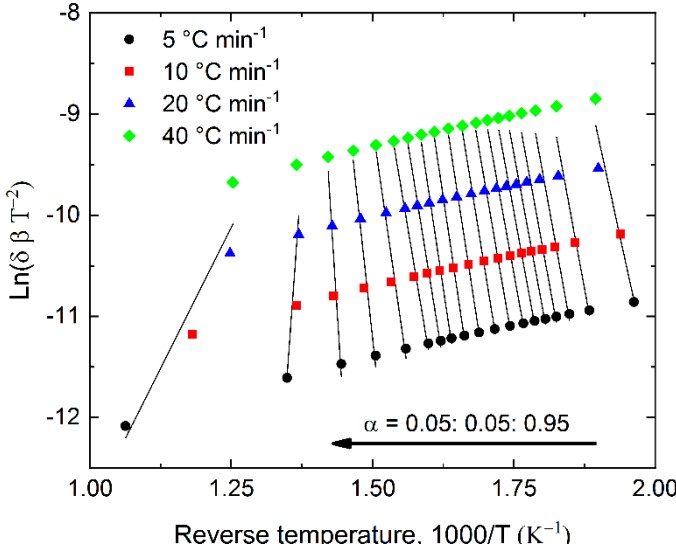

**Figure 4.** Exemplary linear fit plot using Kissinger–Akahira–Sunose (KAS) method to determine the activation energy for BSG.

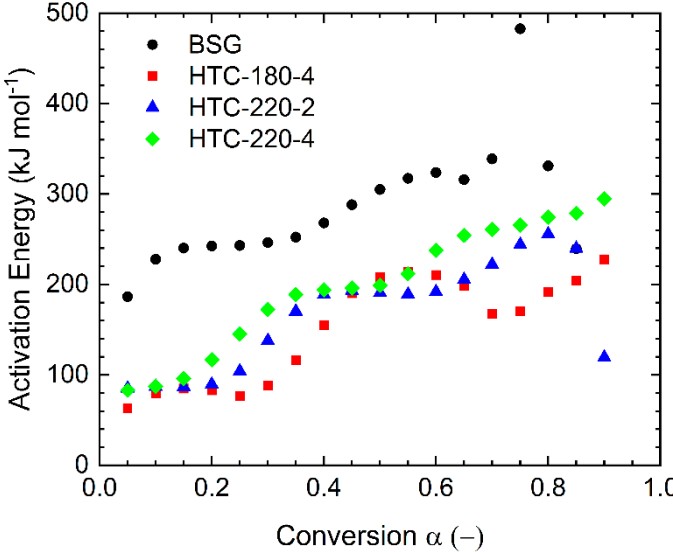

**Figure 5.** Calculated activation energies for investigated materials.

The kinetic parameters are summarized in Table A2 (Appendix A). The apparent activation energy calculated for BSG (one-step process), for the conversion range between 5% and 85% resulted in values from 187–338 kJ mol$^{-1}$, with an average of 285 kJ mol$^{-1}$. Obtained values were similar to coffee ground residues (~190–353 kJ mol$^{-1}$) presented in the work of Mašek et al. [38], but they were a bit higher than those obtained for other biomasses, e.g., tobacco plant waste (118–257 kJ mol$^{-1}$) [51] and rice husk (221–229 kJ mol$^{-1}$) [52]. The activation energies calculated for the pyrolysis of the hydrochars were in the ranges 63–214 kJ mol$^{-1}$, 85–256 kJ mol$^{-1}$, and 83–279 kJ mol$^{-1}$ for HTC-180-4, HTC-220-2, and HTC-220-4, respectively. Such results led to the presumption that the activation energies for hydrochars produced from BSG were lower than the activation energy for the parent feedstock. Recently published research about pyrolysis kinetics of hydrochar by Hameed et al. [48] showed activation energy (91–163 kJ mol$^{-1}$) for hydrochar produced from Karanji fruit hulls at 200 °C during 5 h residence time. Their values for the hydrochars were higher than the initial biomass (28–93 kJ mol$^{-1}$). However, their values for the parent biomass seem to be very low. The average activation energies for hydrochars produced from sawdust presented by Li et al. [34] were in the

range of 150–195 kJ mol$^{-1}$, which gives similar results as those obtained in this study 147, 170, and 188 kJ mol$^{-1}$ for HTC-180-4, HTC-220-2, and HTC-220-4, respectively. The differences can arise due to the use of different biomasses, heating rates, and final temperatures during the thermogravimetric experiments as well as the complexity of the pyrolysis process.

The lower activation energy for hydrochars may confirm the fact that the thermochemical conversion occurring in wet conditions (HTC or wet torrefaction) proceeds according to a different mechanism than in dry processing (torrefaction and pyrolysis). Hydrochars also were characterized by a faster mass loss in the initial stage of pyrolysis at rather low temperatures (up to 300 °C) related to the DTG$_{1*}$ peaks. The much lower activation energy for hydrochars (below 90 kJ mol$^{-1}$) at the initial stages of conversion, comprising about 30% compared to those of biomass (187 kJ mol$^{-1}$) may have been caused by the absence of hemicellulose in hydrochars which was hydrolyzed during HTC. A characteristic bulge in the activation energy plot appears for all materials in the conversion range from 0.4–0.7. The conversion 0.4 corresponds to a temperature of around 350 °C, which is related to the decomposition of cellulose (peak DTG$_2$). For HTC-180-4, the convexity was visible with the highest intensity; it could have been caused by the high cellulose content, which had not been converted because of the low temperatures in the HTC process. For two other hydrochars which were produced at 220 °C, the bulge decreased, which confirms that cellulose had already been converted to a large extent during the HTC process. Alternatively, lower activation energies of hydrochars pyrolysis may also be due to the chemical rearrangement of the biomass structure during the hydrothermal carbonization process, where new material is formed during the polymerization of intermediate products such as 5-HMF [9,16].

The final summary of the pyrolysis kinetics requires the pre-exponential factor (A) to be specified, which explains reaction chemistry. In this study, the Coats–Redfern (CR) method was used to determine the reaction mechanisms of the thermal decomposition of BSG and derived hydrochars. The conversion ($\alpha$) was divided into two ranges: (I) $\alpha = 0.1-0.4$ and (II) $\alpha = 0.45-0.85$ due to the characteristic shape of curves representing the activation energies in Figure 5. Then different reaction mechanisms represented as the integral function g ($\alpha$) were used to calculate the activation energy according to Equation (10). The exemplary calculation for BSG is shown in Table A3 (Appendix A). All investigated reaction mechanisms showed a very good linear correlation ($R^2 \geq 0.95$) in the range I. Nevertheless, the activation energies for these models did not coincide with the values obtained with the KAS method (245.76 kJ mol$^{-1}$). The closest value of E$_A$ (180.42 kJ mol$^{-1}$) was achieved for three-dimensional diffusion (D3); however, this value was lower ca. 26%. Therefore, this mechanism cannot be unambiguously assigned. In the range II, the linear fit of the analyzed reaction mechanisms was not as accurate as the previous range. Only a few mechanisms showed a good correlation. However, none of these mechanisms were close to the expected values of activation energy obtained from the KAS model. Similar results were obtained for hydrochars. The results showed that it is not possible to determine which reaction mechanism is appropriate in the given conversions ranges.

As previously mentioned and confirmed by the results, pyrolysis is a highly complex process, and its exact mechanism is not clearly defined yet. Due to the lack of knowledge about the proper reaction mechanisms that occurred during the thermal decomposition, the calculation of the pre-exponential factor (A) using the KAS or CR method is not feasible. Herein, therefore, the A factor was estimated using a simplified method based on Equation (11), which uses the temperature corresponding to the highest DTG peak. The values of parameter A for BSG pyrolysis were in the range of $10^{12}$–$10^{29}$ s$^{-1}$ for conversions up to 70%. For higher conversion levels, the pre-exponential factor increased very rapidly, up to $10^{166}$ s$^{-1}$. Such tremendous difference may be connected with an instability of the applied calculation method for the higher conversion range (low $R^2$ value = 0.59, Table A2). For hydrochars, the values of the pre-exponential parameter were in the range of $10^2$–$10^{14}$ s$^{-1}$ for HTC-180-4 and $10^4$–$10^{21}$ s$^{-1}$ for both hydrochars produced at 220 °C. In the literature, different ranges of values of pre-exponential factors for different biomasses can be found, i.e., $10^{10}$–$10^{15}$ s$^{-1}$ for bulrush [50], $10^7$–$10^{12}$ s$^{-1}$ for rice straw, $10^3$–$10^{21}$ s$^{-1}$ for switchgrass [49], and $10^3$–$10^{10}$ s$^{-1}$ for hydrochar (Karanj

fruit hulls) [48]. The ranges of the activation energy and pre-exponential factor for hydrochars given in the literature are higher than those obtained in this study.

Additionally, both parameters are lower for the hydrochars than for the initial biomass. We suspect that such slight discrepancy in results can be caused by the differences in the structure and composition between materials. Biomass properties are strongly determined by the share of each of the three main structural components, namely cellulose, hemicelluloses, and lignin. Lignin is one of cell wall polymers which plays an important role during the decomposition of lignocellulosic biomass due to high structural heterogeneity. It is built from coniferyl (G-guaiacyl), sinapyl (S-syringyl), and p-coumaryl (H-hydroxyphenyl) alcohols, which create a cross-linked heteropolyphenol structure [53]. The complex structure of lignin results in various depolymerization reactions during thermal decomposition [54], which may influence on the pyrolysis kinetic. Our previous study related to Py-GC-MS analysis of BSG showed decomposition products of the guaiacyl and syringyl type of lignin [33]. In connection with a composition of BSG, a critical role is a high content of proteins which may impact the kinetic. As mentioned in Section 2.3, the decomposition of proteins in TGA takes place between 200–500 °C. In consequence, the activation energy for BSG may be higher than hydrochars while during HTC proteins hydrolyze to amino acids and then reacts with carbohydrates via Maillard-type reaction [13,55,56]. Besides the proteins content BSG contains cellulose where hydrogen bonds additionally stabilize the single chains of cellulose connected through C–O–C glycosidic bonds. It results in cellulose having high thermal stability, with high activation energies for the biomasses rich in cellulose. In the case of the hydrochars, the kinetic parameters cannot be assigned that straightforwardly to the structural bio-components, due to the chemical mechanisms involved in the hydrothermal conversion. Firstly, during HTC, carbohydrates (first hemicelluloses and then cellulose) are hydrolyzed in consequence producing sugar-like compounds. The sugars dissolved in the reaction environment, undergo further reactions, like water elimination (dehydration) and polymerization [10]. The formed structure can be described as a complexed polymer from mostly furfural monomers [57], which lacks the ability to form hydrogen bonds and stabilize its single chains. With the increase of HTC temperature (and optionally with further thermal treatment), the crosslinking between the single pseudo-furfural chains start to occur, which have a stronger stabilization of the chains, (even much stronger than hydrogen bonds). Lack of hydrogen bonds in the structure of low-temperature hydrochars may lead to melting and evaporation during pyrolysis. Decomposition of less stable parts of hydrochars or perhaps evaporation, which is visible in Figure 3 (peak $DTG_{1*}$) led to lower activation energies for hydrochars compared to biomass.

It should be mentioned that the Arrhenius equation was initially developed for liquids (dissolved electrolytes and sugar cane) [58,59] which have known molar mass, not for a solid-state reaction. Therefore, activation energies are given in molar quantities. The same model equation was adapted for solids [60–62]; however, in case of complex structures like biomass or hydrochars, the molar masses are unknown. One possibility to come out of this dilemma is calculating an activation temperature ($T_A$, Equation (12)) instead of a molar activation energy. The activation temperatures are presented in Table A2 (Appendix A).

## 2.5. Heat Flow during Pyrolysis Reaction

Differential scanning calorimetry (DSC) curves indicating overall thermal effects occurring during pyrolysis of BSG and hydrochars are shown in Figure 6. BSG and all produced hydrochars showed positive heat flow from the sample, which indicated exothermic reactions occurring during the thermal degradation process. In the case of the decomposition of model substances, the DSC peaks can be related to mass loss in DTG [63]. However, for the decomposition of BSG, which contain hemicellulose, cellulose, lignin, and proteins, the thermal effect could not be directly connected with the DTG peaks (Figure 3B). It means that there is a strong interaction between different structural components. The DSC curve for HTC-180-4 is more similar to BSG than other hydrochars. Both materials have a characteristic peak, which achieved maximum heat flow at 575 °C and 675 °C for BSG and HTC-180-4,

respectively. The similarity results from the lack of cellulose conversion during low-temperature HTC. Despite the conversion of hemicellulose, hydrochar produced at 180 °C resembled the parent feedstock with an initiated phase of conversion into a hydrochar with the relatively higher content of cellulose as a consequence of the mass loss during HTC and the stability of cellulose at low-temperature HTC. Yang et al. [63] reported that decomposition of cellulose at a temperature higher than 400 °C becomes exothermic, and exceeding 650 °C the exothermicity of the process sharply increases. This may explain the high exothermic peaks at 575 °C for BSG and 675 °C for HTC-180-4. In contrast, both hydrochars obtained at 220 °C showed a local minimum in the DSC curves in the same temperature range. The variation between these materials could be associated with higher carbonization degree related to the conversion of cellulose at a higher severity of HTC and thus the drop of the reaction exothermicity at temperatures above 400 °C. In the same way, it may influence the presence of aromatic products from a Maillard-type reaction created during HTC. The exothermic effect of BSG pyrolysis was also observed by Ferraz et al. [64]. Subject literature reports that pyrolysis at a small scale (mg) is endothermic [65–68]; however, the exothermic effect of biomass pyrolysis is also reported [69–71]. The final thermal effect of thermal degradation strongly depends on the composition of the investigated material.

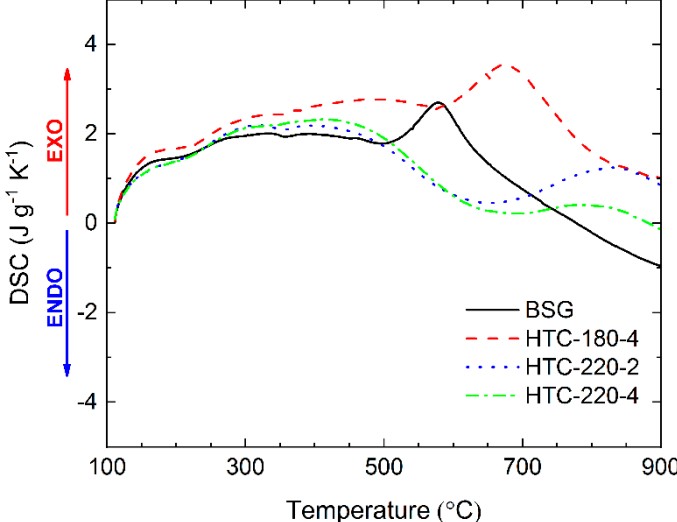

**Figure 6.** Heat flow during pyrolysis of BSG and hydrochars at a heating rate 10 °C min$^{-1}$.

## 3. Materials and Methods

### 3.1. Materials

The brewer's spent grains (BSG) used as feedstock in this research were delivered by the local brewery Hoepfner (Karlsruhe, Germany). Oven-dried (at 105 °C) over 24 h BSG was ground to a particle size below 200 μm for further TGA measurements. Three hydrochars were produced using wet BSG (78 wt.% moisture) at different process conditions: (1) 180 °C, 4 h residence time; (2) 220 °C, 2 h residence time; and (3) 220 °C and 4 h residence time. Obtained hydrochars were named as HTC-180-4, HTC-220-2, and HTC-220-4, respectively. The hydrochars were dried overnight at 105 °C, and were then ground to a particle size below 200 μm. More detailed information about feedstock preparation and hydrochars production was described elsewhere [21].

### 3.2. Pyrolysis Procedure

A series of pyrolysis experiments were conducted using the previously dried BSG and hydrochars. For this purpose, 40 mg of sample was used in order to obtain enough material (i.e., pyrochar) for consecutive analyses. Pyrolysis with a heating rate of 10 °C min$^{-1}$ was performed for the BSG and hydrochars produced at different conditions using a Netzsch STA Jupiter 449 F5 (Selb, Germany) TGA

instrument. The samples were heated up from ambient temperature to 300, 500, 700, and 900 °C and held for 10 min at the desired temperature. Nitrogen gas with a flow of 70 mL min$^{-1}$ was used to provide an inert atmosphere during measurements. The pyrolysis yield, as well as the elemental composition of produced pyrochars, were analyzed.

TGA–DSC measurements were performed in the same instrument as the pyrolysis. Around 5 mg of sample was evenly spread in the crucible to reduce mass and heat transfer limitations [49]. The tested sample was placed in alumina crucibles and heated up to 105 °C, and then kept for 10 min in isothermal conditions to remove moisture from the samples. Afterward, samples were heated up to a temperature of 900 °C. The experiments were carried out using four different heating rates, namely 5, 10, 20, and 40 °C min$^{-1}$, to achieve non-isothermal degradation for further kinetic analysis [50,72,73]. Nitrogen gas with a volumetric flow rate of 70 mL min$^{-1}$ was used to provide an inert atmosphere with the TGA-DSC instrument. All measurements were carried out in triplicate in order to assess repeatability of the results.

### 3.3. Proximate and Ultimate Analysis

Elemental analysis of the BSG, hydrochars, and pyrochars was performed on a CHNS analyzer EuroEA, 3000 Series (HEKAtech GmbH, Wegberg, Germany) according to the standard (DIN-51732). The oxygen content was calculated from the difference between the combined mass of measured elements on the dry ash-free basis. The moisture content, volatile matter (VM), and ash content for raw BSG and hydrochars were analyzed according to industrial standard proximate analysis (ASTM D1762-84). The difference between the sum of measured ash and VM contents from 100% is the value of fixed carbon content (FC) [50]. The measurements were made in duplicates, and the average data were reported. Due to the small amount of pyrochars produced, the ash content could not be directly assessed using the above method. Instead, the ash content of pyrochars was calculated, assuming that all ash from the initial substrate stayed within the final product (i.e., char). Also, the VM of the pyrochars was calculated rather than measured, which was based on the difference between the pyrochar sample mass and the reference char mass obtained at 900 °C in TGA. The higher heating value (HHV, MJ kg$^{-1}$) was calculated based on the elemental composition using the Channiwala and Parikh equation (Equation (1)) [74]. The formula is widely used because of its high accuracy. Additionally, it may be used for a wide range of fuels, including solid, liquid, and gaseous fuels.

$$HHV = 0.3491C + 1.1783H + 0.1005S - 0.1034O - 0.0151N - 0.021A, \tag{1}$$

where, *C*, *H*, *S*, *O*, *N*, and *A* refer to carbon, hydrogen, sulfur, oxygen, nitrogen, and ash content, respectively (in wt.%).

### 3.4. Mathematical Model for Pyrolysis Kinetics

According to the literature of the subject, a one-step, global model is an efficient method to describe the non-isothermal pyrolysis kinetics and to perform preliminary analysis [38,48,50]. The model assumes that the process occurs as a single reaction, where the feedstock is converted to char (solid residue) with the release of volatiles (gas and bio-oil) as it is shown in Equation (2) [38].

$$Feedstock \xrightarrow{k} Char + Volatiles. \tag{2}$$

The solid material decomposition rate to analyze the TGA data can be described as follows:

$$\frac{d\alpha}{dt} = k(T)\, f(\alpha), \tag{3}$$

where, $f(\alpha)$ is a conversion function which depends on the reaction mechanism, and $d\alpha/dt$ is the conversion rate of the feedstock over the reaction time ($t$). Conversion ($\alpha$) is calculated by the following expression:

$$\alpha = \frac{m_0 - m_t}{m_0 - m_f},\tag{4}$$

where, $m_0$, $m_t$, $m_f$ correspond to initial mass of the feedstock, mass after reaction time t, and final remaining mass after the process, respectively. In Equation (3), $k(T)$ is the reaction rate constant, which depends on the temperature and is described by the Arrhenius equation:

$$k(T) = A\, exp\left(-\frac{E_A}{RT}\right),\tag{5}$$

where $A$ is the pre-exponential factor ($s^{-1}$), $E_A$ is the activation energy ($J\ mol^{-1}$), $R$ is the universal gas constant ($J\ K^{-1}\ mol^{-1}$), and $T$ is temperature ($K$).

In the non-isothermal thermogravimetric analysis, the temperature ($T$) is increasing linearly with the known heating rate ($\beta$) over time ($t$), which is represented as follows:

$$\beta = \frac{dT}{dt}.\tag{6}$$

Subsequently, by substituting Equations (5) and (6) into Equation (3), the differential form of non-isothermal decomposition is obtained:

$$\frac{d\alpha}{dT} = \frac{A}{\beta} exp\left(-\frac{E_A}{RT}\right) f(\alpha).\tag{7}$$

Finally, the integration of Equation (7) for the initial condition $T = T_0$ and $\alpha = 0$, results in the fundamental equation, which is the basis for all kinetic methods to determine the kinetic parameters during the pyrolysis process:

$$\int_0^\alpha \frac{d\alpha}{f(\alpha)} = g(\alpha) = \frac{A}{\beta} \int_{T_0}^T \exp\left(-\frac{E_A}{RT}\right) dT.\tag{8}$$

The expression of the reaction mechanism in the form of the derivative [$f(\alpha)$] or the integral [$g(\alpha)$] can be found elsewhere [75].

### 3.5. Calculation of Pyrolysis Kinetics

Proper implementation of the reaction mechanism is a crucial point to obtain proper kinetic parameters of the conversion reaction. Therefore in the literature, several methods of approach to proper implementation of the reaction mechanism model can be found, i.e., Coats and Redfern's model-fitting method (CR) [60]. Kinetic parameters could also be determined by isoconversional (model-free) methods established by Kissinger [61], Friedman [76], Flynn–Wall–Ozawa (FWO) [62,77], and Kissinger–Akahira–Sunose (KAS) [78]. Studies on the comparison of the kinetic parameters from the different model-free methods have shown that the results between approaches are very similar [38,50,79]. The model-free methods allow omitting the choice of the reaction mechanism, and in this way, eliminate the error associated with the implementation of an inappropriate mechanism [75].

The KAS method was used to characterize the pyrolysis kinetic parameter of brewer's spent grains and its hydrochars. The kinetics parameter calculations were conducted in the temperature range from 105–800 °C for different heating rates (5, 10, 20, and 40 °C min$^{-1}$) with three replicates (A, B, C). The applied calculation technique is expressed as follows:

$$Ln\left(\frac{\beta}{T_\alpha^2}\delta\right) = Ln\left(\frac{A_\alpha R}{E_{A\alpha} g(\alpha)}\delta\right) - \frac{E_{A\alpha}}{T_\alpha R},\tag{9}$$

where $\beta$, $T_\alpha$, $A_\alpha$, $E_{A\alpha}$, and $\delta$ refer to heating rate (K min$^{-1}$), temperature of desired conversion (K), pre-exponential factor (min$^{-1}$), apparent activation energy (J mol$^{-1}$) for a fixed degree of conversion $\alpha$ [$-$], and unit correction factor (K$^{-1}$ min$^{-1}$), respectively. The apparent activation energy for a selected degree of conversion was calculated as the slope from plotting Ln($\beta/T_\alpha{}^2$) versus $1/T_\alpha$. The knowledge of reaction mechanism represented as an integrated form g($\alpha$) in Equation (9) is required to calculate the pre-exponential factor A using the KAS model. One of the methods used for the determination of a reaction mechanism during the thermal decomposition is Coats-Redfern (CR) method [60]. The method is presented as follow:

$$Ln\left(\frac{g(\alpha)}{T_\alpha^2}\delta\right) = Ln\left(\frac{A_\alpha R}{\beta E_{A\alpha}}\delta\right) - \frac{E_{A\alpha}}{T_\alpha R}, \tag{10}$$

where function $g(\alpha)$ assumes a form assigned to the different reaction mechanisms [75]. Plotting the Ln($g(\alpha)/T_\alpha{}^2$) versus $1/T_\alpha$ allows calculating the activation energy $E_{A\alpha}$ from the slope. The activation energy obtained using the CR method for different reaction mechanisms should be compared to activation energy obtained from a different model (herein KAS). When the activation energy for the specific reaction mechanism is close to the activation energy obtained using KAS method, it can be assumed that the thermal decomposition undergoes according to this reaction mechanism [50].

Unfortunately, any of the tested reaction mechanisms have not shown a good match. Therefore, the pre-exponential factors ($A$) were calculated using a simplified method based on [80]:

$$A = \frac{\beta E_A exp\left(\frac{E_A}{RT_m}\right)}{RT_m^2}, \tag{11}$$

where $T_m$ is a DTG peak temperature [K].

The average temperature from three replicates for each degree of conversion could be used for the calculation of the activation energy. However, for a larger set of experimental data, a permutation was used in order to obtain more precise and reliable results. In this particular case, the permutation with repetitions should be used, where elements $k = 3$ (3-replicates: A, B, C) and the number of n-tuples = 4 (4 different heating rates). It resulted in $3^4 = 81$ configurations (AAAA, ABAA, ... , CCCC) for the calculation. The results obtained with the permutation method were shown as average values.

Additionally, the activation temperature ($T_A$) (K) was calculated from the following equation:

$$T_A = \frac{E_A}{R}, \tag{12}$$

where, $E_A$ is activation energy (J mol$^{-1}$), and $R$ is the universal gas constant (J K$^{-1}$ mol$^{-1}$).

## 4. Conclusions

A non-isothermal thermogravimetric analysis with four heating rates (5, 10, 20, 40 °C min$^{-1}$) was performed. The Kissinger–Akahira–Sunose method was used to determine apparent activation energies ($E_A$), which were 285, 147, 170, and 188 kJ mol$^{-1}$ for BSG, HTC-180-4, HTC-220-2, and HTC-220-4, respectively. An attempt was made to match the reaction mechanisms during pyrolysis using the Coats–Redfern method, but none of the mechanisms showed a right approach. From DTG curves it was found that hemicellulose was already converted in the HTC process at temperatures of 180 °C, while cellulose was mostly unreacted and the decomposition of cellulose was only observed for hydrochars produced at 220 °C. Additionally, a series of pyrolysis experiments at 300, 500, 700, and 900 °C were conducted using a TGA instrument. The effect of the HTC process condition on the pyrochars yield was observed. Generally, the pyrolysis yield was higher for hydrochars than for its parent biomass. Increasing the severity of the HTC conditions results in higher pyrolysis yields at 900 °C, which were 22.0%, 28.3%, 37.4%, 45.8% for BSG, HTC-180-4, HTC-220-2, and HTC-220-4, respectively. The additional process will also reduce the overall yield of char production. However,

the coupling of both processes may bring benefit to improve the quality of obtained pyrochars (i.e., physicochemical properties). Additionally, in some particular cases (i.e., high moist feedstock and selected operating conditions) this solution may reduce the energy demand of the whole process by saving the energy necessary for drying. More studies related to preprocessing (i.e., dewatering and drying) and pyrolysis of this material are essential to estimate the economic aspect. Unfortunately, higher amounts of hydrochars are required for further studies, to achieve reliable mass and energy balance for economic evaluation. Findings in this study showed that the HTC process preceding pyrolysis can extend the range of wet biomasses for bio-refinery purposes due to lower pyrolysis activation energies for hydrochars produced from brewer's spent grains.

**Author Contributions:** Conceptualization, M.P.O. and A.K.; methodology, M.P.O. and P.J.A.; software, M.P.O. and P.A.M.; validation, M.P.O., F.R., and A.K.; formal analysis, M.P.O. and P.A.M.; investigation, M.P.O. and P.J.A.; resources, M.P.O.; data curation, M.P.O.; writing—original draft preparation, M.P.O.; writing—review and editing, M.P.O., F.R., and A.K.; visualization, M.P.O. and P.A.M.; supervision, A.K.; project administration, M.P.O. and A.K.; funding acquisition, A.K.

**Funding:** This project has received funding from the European Union's Horizon 2020 research and innovation program under the Marie Skłodowska-Curie grant agreement No 721991.

**Conflicts of Interest:** The authors declare no conflict of interest.

## Appendix A

**Table A1.** Characteristic parameters during thermal decomposition of brewer's spent grains and its hydrochars at different heating rates.

| Material | Heating Rate $^\circ$C min$^{-1}$ | $T_1$, $T_{1,*}$ $^\circ$C | DTG$_1$ DTG$_{1,*}$ % $^\circ$C$^{-1}$ | T2 $^\circ$C | DTG$_2$ % $^\circ$C$^{-1}$ | $T_3$, $T_{3*}$ $^\circ$C | DTG$_3$ % $^\circ$C$^{-1}$ | Final Residue wt.% |
|---|---|---|---|---|---|---|---|---|
| BSG | 5 | 281.2 | −0.687 | 340.5 | −0.523 | 367.5 | −0.198 | 18.98 |
| | 10 | 289.4 | −0.736 | 349.6 | −0.503 | 378.3 | −0.202 | 19.24 |
| | 20 | 298.6 | −0.776 | 357.9 | −0.515 | 388.3 | −0.211 | 20.99 |
| | 40 | 304.4 | −0.887 | 358.3 | −0.630 | 402.5 | −0.228 | 20.44 |
| | average | 293.4 | −0.771 | 351.6 | −0.543 | 384.2 | −0.210 | 19.91 |
| HTC-180-4 | 5 | 211.2 | −0.285 | 339.8 | −0.592 | 404.2 | −0.169 | 24.97 |
| | 10 | 223.9 | −0.267 | 349.4 | −0.556 | 412.1 | −0.165 | 29.78 |
| | 20 | 236.0 | −0.320 | 358.6 | −0.543 | 422.3 | −0.172 | 29.37 |
| | 40 | 251.6 | −0.317 | 362.2 | −0.633 | 419.9 | −0.183 | 29.08 |
| | average | 230.7 | −0.297 | 352.5 | −0.581 | 414.6 | −0.172 | 28.30 |
| HTC-220-2 | 5 | 213.0 | −0.268 | 340.1 | −0.411 | 410.0 | −0.168 | 38.71 |
| | 10 | 225.8 | −0.276 | 353.3 | −0.414 | 415.8 | −0.162 | 39.54 |
| | 20 | 243.3 | −0.325 | 359.5 | −0.370 | 427.0 | −0.167 | 41.32 |
| | 40 | 269.1 | −0.326 | 378.3 | −0.426 | 423.7 | −0.155 | 43.17 |
| | average | 237.8 | −0.298 | 357.8 | −0.405 | 419.1 | −0.163 | 40.68 |
| HTC-220-4 | 5 | 209.2 | −0.171 | 342.8 | −0.298 | 410.3 | −0.193 | 44.62 |
| | 10 | 221.3 | −0.175 | 353.9 | −0.302 | 427.4 | −0.194 | 45.60 |
| | 20 | 243.9 | −0.216 | 361.8 | −0.279 | 436.4 | −0.192 | 46.13 |
| | 40 | 269.4 | −0.236 | 378.9 | −0.340 | 441.1 | −0.176 | 46.66 |
| | average | 236.0 | −0.199 | 359.4 | −0.305 | 428.8 | −0.189 | 45.75 |

**Table A2.** Summary of the kinetic parameters for brewer's spent grains and its hydrochars.

| $\alpha-$ | $E_A$ kJ mol$^{-1}$ | $R^{2-}$ | A s$^{-1}$ | $T_A$ K $\times 10^3$ | $E_A$ kJ mol$^{-1}$ | $R^{2-}$ | A s$^{-1}$ | $T_A$ K $\times 10^3$ |
|---|---|---|---|---|---|---|---|---|
| **Material** | | **BSG** | | | | **HTC-180-4** | | |
| 0.05 | 186.50 | 0.81 | $1.17 \times 10^{12}$ | 22.43 | 62.89 | 0.79 | $9.05 \times 10^{2}$ | 7.56 |
| 0.10 | 227.85 | 0.88 | $1.66 \times 10^{18}$ | 27.41 | 79.24 | 0.87 | $2.72 \times 10^{4}$ | 9.53 |
| 0.15 | 240.19 | 0.93 | $1.34 \times 10^{20}$ | 28.89 | 84.80 | 0.91 | $8.50 \times 10^{4}$ | 10.20 |
| 0.20 | 242.52 | 0.95 | $3.54 \times 10^{20}$ | 29.17 | 82.86 | 0.89 | $5.68 \times 10^{4}$ | 9.97 |
| 0.25 | 243.15 | 0.97 | $4.64 \times 10^{20}$ | 29.25 | 76.83 | 0.79 | $1.57 \times 10^{4}$ | 9.24 |
| 0.30 | 246.34 | 0.98 | $9.45 \times 10^{20}$ | 29.63 | 88.18 | 0.74 | $1.37 \times 10^{5}$ | 10.61 |
| 0.35 | 252.30 | 0.98 | $3.44 \times 10^{21}$ | 30.35 | 116.50 | 0.76 | $3.77 \times 10^{7}$ | 14.01 |
| 0.40 | 267.98 | 0.98 | $9.68 \times 10^{22}$ | 32.23 | 154.98 | 0.81 | $7.6 \times 10^{10}$ | 18.64 |
| 0.45 | 288.12 | 0.98 | $8.09 \times 10^{24}$ | 34.65 | 190.25 | 0.87 | $9.41 \times 10^{13}$ | 22.88 |
| 0.50 | 305.07 | 0.98 | $3.14 \times 10^{26}$ | 36.69 | 208.11 | 0.90 | $3.57 \times 10^{15}$ | 25.03 |
| 0.55 | 317.36 | 0.98 | $3.92 \times 10^{27}$ | 38.17 | 213.69 | 0.93 | $1.12 \times 10^{16}$ | 25.70 |
| 0.60 | 323.89 | 0.98 | $1.60 \times 10^{28}$ | 38.96 | 210.42 | 0.93 | $6.01 \times 10^{15}$ | 25.31 |
| 0.65 | 315.99 | 0.98 | $2.96 \times 10^{27}$ | 38.01 | 198.37 | 0.91 | $5.78 \times 10^{14}$ | 23.86 |
| 0.70 | 338.78 | 0.96 | $3.64 \times 10^{29}$ | 40.75 | 167.24 | 0.77 | $1.15 \times 10^{12}$ | 20.12 |
| 0.75 | 482.71 | 0.82 | $4.30 \times 10^{43}$ | 58.06 | 170.34 | 0.72 | $1.38 \times 10^{12}$ | 20.49 |
| 0.80 | 331.10 | 0.59 | $3.34 \times 10^{69}$ | 39.82 | 191.85 | 0.74 | $5.42 \times 10^{13}$ | 23.08 |
| 0.85 | 239.62 | 0.59 | $1.72 \times 10^{166}$ | 28.82 | 204.37 | 0.77 | $2.15 \times 10^{14}$ | 24.58 |
| average | 285.26 | 0.90 | – | 34.31 | 147.11 | 0.83 | – | 17.69 |
| **Material** | | **HTC-220-2** | | | | **HTC-220-4** | | |
| 0.05 | 85.19 | 0.98 | $9.98 \times 10^{4}$ | 10.25 | 78.39 | 0.99 | $4.25 \times 10^{4}$ | 9.43 |
| 0.10 | 87.22 | 0.99 | $1.30 \times 10^{5}$ | 10.49 | 82.98 | 0.99 | $1.10 \times 10^{5}$ | 9.98 |
| 0.15 | 86.77 | 0.99 | $1.15 \times 10^{5}$ | 10.44 | 91.69 | 0.98 | $6.47 \times 10^{5}$ | 11.03 |
| 0.20 | 89.62 | 0.98 | $2.06 \times 10^{5}$ | 10.78 | 112.46 | 0.95 | $4.14 \times 10^{7}$ | 13.53 |
| 0.25 | 104.21 | 0.96 | $4.43 \times 10^{6}$ | 12.53 | 142.34 | 0.91 | $1.56 \times 10^{10}$ | 17.12 |
| 0.30 | 137.92 | 0.93 | $5.40 \times 10^{9}$ | 16.59 | 167.08 | 0.92 | $2.07 \times 10^{12}$ | 20.10 |
| 0.35 | 170.20 | 0.93 | $3.53 \times 10^{12}$ | 20.47 | 183.23 | 0.95 | $5.01 \times 10^{13}$ | 22.04 |
| 0.40 | 188.88 | 0.96 | $1.14 \times 10^{14}$ | 22.72 | 189.07 | 0.97 | $1.60 \times 10^{14}$ | 22.74 |
| 0.45 | 193.33 | 0.98 | $2.05 \times 10^{14}$ | 23.25 | 191.74 | 0.98 | $2.75 \times 10^{14}$ | 23.06 |
| 0.50 | 190.65 | 0.98 | $1.09 \times 10^{14}$ | 22.93 | 194.98 | 0.97 | $5.26 \times 10^{14}$ | 23.45 |
| 0.55 | 189.12 | 0.98 | $7.75 \times 10^{13}$ | 22.75 | 208.77 | 0.94 | $7.90 \times 10^{15}$ | 25.11 |
| 0.60 | 191.91 | 0.98 | $1.39 \times 10^{14}$ | 23.08 | 235.17 | 0.88 | $1.40 \times 10^{18}$ | 28.29 |
| 0.65 | 205.44 | 0.89 | $5.49 \times 10^{15}$ | 24.71 | 251.28 | 0.89 | $3.31 \times 10^{19}$ | 30.22 |
| 0.70 | 222.05 | 0.76 | $3.69 \times 10^{17}$ | 26.71 | 258.15 | 0.91 | $1.29 \times 10^{20}$ | 31.05 |
| 0.75 | 244.16 | 0.74 | $5.03 \times 10^{19}$ | 29.37 | 263.15 | 0.92 | $3.51 \times 10^{20}$ | 31.65 |
| 0.80 | 255.68 | 0.73 | $1.31 \times 10^{21}$ | 30.75 | 273.82 | 0.92 | $2.88 \times 10^{21}$ | 32.93 |
| 0.85 | 239.95 | 0.63 | $2.18 \times 10^{21}$ | 28.86 | 279.40 | 0.90 | $8.81 \times 10^{21}$ | 33.61 |
| average | 169.55 | 0.90 | – | 20.39 | 188.45 | 0.94 | – | 22.67 |

**Table A3.** The activation energy for BSG calculated using Coats–Redfern method and several different reaction mechanisms, The reaction mechanisms and their equations are adapted from [50,81,82].

| Reaction Mechanism | Integral Form, g($\alpha$) | Range I $\alpha = 0.10-0.40$ | | Range II $\alpha = 0.45-0.85$ | |
|---|---|---|---|---|---|
| | | $E_A$ kJ mol$^{-1}$ | $R^{2-}$ | $E_A$ kJ mol$^{-1}$ | $R^{2-}$ |
| **Reaction Order** | | | | | |
| Zero order F0 | $\alpha$ | 77.58 | 0.98 | 9.39 | 0.74 |
| First order F1 | $-\ln(1-\alpha)$ | 89.76 | 0.98 | 26.11 | 0.95 |
| Second order F2 | $[1/(1-\alpha)]-1$ | 103.21 | 0.99 | 51.11 | 0.99 |
| Third order F3 | $[1/(1-\alpha)^2]-1$ | 117.93 | 0.99 | 83.16 | 0.99 |
| **Diffusion** | | | | | |
| One-dimensional diffusion D1 | $\alpha^2$ | 164.46 | 0.98 | 29.44 | 0.88 |
| Two-dimensional diffusion D2 | $(1-\alpha)\ln(1-\alpha)+\alpha$ | 172.17 | 0.99 | 38.12 | 0.92 |
| Three-dimensional diffusion D3 | $[1-(1-\alpha)^{1/3}]^2$ | 180.42 | 0.99 | 49.91 | 0.95 |
| Ginstling–Brouns D4 | $1-(2\alpha/3)-(1-\alpha)^{2/3}$ | 174.92 | 0.99 | 41.98 | 0.93 |
| **Geometrical Contraction Models** | | | | | |
| Contracting area (cylinder) R2 | $1-(1-\alpha)^{1/2}$ | 83.51 | 0.99 | 10.76 | 0.86 |
| Contracting volume (sphere) R3 | $1-(1-\alpha)^{1/3}$ | 85.56 | 0.99 | 13.60 | 0.90 |
| **Nucleation Models** | | | | | |
| Power law P1 | $\alpha^{1/4}$ | 12.41 | 0.95 | $-5.65$ | 0.93 |
| Power law P2 | $\alpha^{1/3}$ | 19.65 | 0.97 | $-3.98$ | 0.79 |
| Power law P3 | $\alpha^{1/2}$ | 34.13 | 0.98 | $-0.64$ | 0.05 |
| Power law P4 | $\alpha^{3/2}$ | 48.62 | 0.98 | 2.70 | 0.34 |
| Avrami–Erofeev (m = 2) A2 | $[-\ln(1-\alpha)]^{1/2}$ | 40.22 | 0.99 | 7.72 | 0.85 |
| Avrami–Erofeev (m = 3) A3 | $[-\ln(1-\alpha)]^{1/3}$ | 23.71 | 0.98 | 1.59 | 0.34 |
| Avrami–Erofeev (m = 4) A4 | $[-\ln(1-\alpha)]^{1/4}$ | 15.46 | 0.97 | $-1.47$ | 0.42 |
| **Average $E_A$ Obtained Using KAS Method** | | 245.76 | – | 326.96 | – |

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
