# Peer review of "Pyrolysis Kinetics of Hydrochars Produced from Brewer’s Spent Grains"

_catalysts, doi:10.3390/catal9070625_

Round 1
Reviewer 1 Report
The manuscript describes the pyrolysis of brewer's grains by several react5ion conditions. The manuscript is well summarized and at the acceptable status. But The reviewer want to see the following queries,
1) The results obtained are specific for Brewer's spent grains? The component is much complicated and can you add some comments?
2) In Table 2, can you show how to calculate char yield?
3) As Figure 5, activation energy is dependent on the conversion. in case of BSG, it looks two stages and also to the other materials it looks same. Can you say some comments?
4) on page 11, in last paragraph, what is 'fluid'? I am not sure what means?
Reviewer 2 Report
Dear Authors
I reaaly enjoy by reading and reviewing this paper. The introduction was excellent written, and th figure1 was very informative. I like your approach. The results were clearly presented and well discussed. All the figures were very supportive to the document.

Reviewer 3 Report
In this work kinetic parameters of pyrolytic degradation of Brewer’s Spent Grains (BSG) and derived hydrochars formed under different conditions have been determined by using a model-free method of Kissinger-Akahira-Sunose. Authors properly calculated the value of kinetic parameter E, whereby the Eq. 10, used for A calculation, is not recommended by ICTAC Kinetics Committee as it is over-simplified - it is based on single parameter 'DTG peak temperature'. The full kinetic analysis should also include determination of the f(alpha) function (Eq. 3) - why these calculations were not performed that are essential to determine the mechanism of decompostion? When f(alpha) function is known, one can also predict the system's behaviour in extrapolated time-, temperature and degree of conversion range that may be very useful for process optimization.
Round 2
Reviewer 3 Report
The revised manuscript can be published as it stands.